# The Association between Dynamic Changes in Serum Presepsin Levels and Mortality in Immunocompromised Patients with Sepsis: A Prospective Cohort Study

**DOI:** 10.3390/diagnostics11010060

**Published:** 2021-01-02

**Authors:** Jongmin Lee, Seohyun Kim, Kyung Hoon Kim, Na Ri Jeong, Seok Chan Kim, Eun-Jee Oh

**Affiliations:** 1Division of Pulmonary, Allergy and Critical Care Medicine, Department of Internal Medicine, Seoul St. Mary’s Hospital, College of Medicine, The Catholic University of Korea, Seoul 06591, Korea; dibs03@gmail.com (J.L.); shyune318@gmail.com (S.K.); nazir0618@gmail.com (N.R.J.); 2Division of Pulmonary, Allergy and Critical Care Medicine, Department of Internal Medicine, Incheon St. Mary’s Hospital, College of Medicine, The Catholic University of Korea, Incheon 21431, Korea; pulmohoon@gmail.com; 3Department of Laboratory Medicine, Seoul St. Mary’s Hospital, College of Medicine, The Catholic University of Korea, Seoul 06591, Korea

**Keywords:** sepsis, immunocompromised, presepsin

## Abstract

Presepsin is a subtype of soluble CD14 that is increased in the blood of septic patients. We investigated the role of dynamic changes in serum presepsin levels in critically ill, immunocompromised patients with sepsis. This is a prospective cohort study that included 119 adult patients admitted to the intensive care unit (ICU). Presepsin level was measured on day 1 and day 3 after ICU admission. The primary outcome was in-hospital mortality. In immunocompromised patients, presepsin levels on day 1 were higher in patients with sepsis than those in patients without sepsis. The area under the curve (AUC) of presepsin for diagnosing sepsis in immunocompromised patients was 0.87, which was comparable with that of procalcitonin (AUC, 0.892). Presepsin levels on day 3 were higher in patients who died in the hospital than in those who survived. In immunocompromised patients who died in the hospital, presepsin levels on day 3 were significantly higher than those on day 1. In the multivariate analysis, ΔPresepsin+ alone was independently correlated with in-hospital mortality in immunocompromised patients. These findings suggest that dynamic changes in presepsin levels between day 1 and day 3 are associated with in-hospital mortality in patients with sepsis, especially in immunocompromised patients.

## 1. Introduction

Sepsis is a syndrome that results in life-threatening organ dysfunction caused by a dysregulated host response to infection [1]. To evaluate the prognosis of sepsis, the Acute Physiology and Chronic Health Evaluation (APACHE) II score and Simplified Acute Physiology Score 3 (SAPS 3) are usually used. However, assessment of prognosis using these scoring systems is intricate and time-consuming. Therefore, a simple biomarker for evaluating the prognosis of sepsis is needed.

Presepsin is a truncated N-terminal fragment of soluble CD14, which is released into the blood upon the activation of monocytes during an inflammatory response [2]. CD14 is a 53–55 kDa glycosylphosphatidylinositol (GPI)-anchored protein lacking a cytoplasmic domain. CD14 is expressed on most innate immune response cells and exists in two forms, one anchored to the membrane by a GPI (mCD14) or in a circulating soluble form (sCD14) [3]. The soluble CD14 subtype, which is called presepsin, has been found to activate a proinflammatory cascade on encountering microorganisms [4].

The levels of this molecule increase during the early phases of infection; previous studies have stated that presepsin levels rise within 2 h of inflammation onset, which is earlier than that reported for C-reactive protein, procalcitonin, and interleukin-6 [5,6]. Recently, presepsin has been used as an early marker for sepsis in emergency departments and intensive care units (ICUs); the sensitivity and specificity of presepsin for diagnosing sepsis is reported to be about 75–80% in most studies [2,7,8,9,10]. Furthermore, studies have previously shown that presepsin levels correlated with the severity or prognosis of sepsis in comparison with other conventional biomarkers [11,12].

Previous studies have reported that plasma presepsin levels are increased in immunocompromised patients with sepsis [13,14,15]. However, few studies have evaluated the prognostic power of presepsin in critically ill, immunocompromised patients with sepsis. We conducted a prospective observational study to assess the prognostic power of plasma presepsin levels at different time points and to evaluate the association between dynamic changes in presepsin levels and the prognosis of immunocompromised patients with sepsis.

## 2. Results

### 2.1. Baseline Characteristics

A total of 119 patients (62 men and 57 women) were included in this study, and 88 (73.9%) of these patients were still in the ICU on day 3. Of the included patients, 58 (48.7%) were immunocompromised. Baseline characteristics of the patients are shown in Table 1. The median age of the patients was 70.0 years (IQR 58.0–79.5), and immunocompromised patients were significantly younger than immunocompetent patients (*p* < 0.001). Sepsis was diagnosed in 40 patients (33.6%), and septic shock was diagnosed in 60 (50.4%) patients. The median initial Sequential Organ Failure Assessment (SOFA) score and mean SAPS 3 on day 1 were 9.0 (6.5–11.5) and 75.5 ± 14.9, respectively, and both scores were significantly higher in immunocompromised patients (*p* = 0.019 for SOFA; *p* = 0.004 for SAPS 3). The median presepsin level at day 1 was 1254.0 pg/mL (730.5–2569.5) and was significantly higher in immunocompromised patients (*p* = 0.024). Among the cases, 53 (44.5%) had hypoxemia requiring invasive ventilation and 77 (64.7%) needed vasopressor support.

### 2.2. Presepsin and Procalcitonin Changes in Sepsis Patients

As shown in Figure 1, we compared presepsin and procalcitonin levels according to the presence of sepsis and immune status. Amongst all patients, the median presepsin level was the highest in patients with septic shock (1766.5 (771.0–2809.5)) and the lowest in patients without septic shock (753.0 (603.5–1092.0)); this difference between patients with and without septic shock was significant (*p* < 0.001). The median presepsin level in patients with sepsis was also significantly higher than that of patients without sepsis (1203.0 (773.0–2484.0) vs. 753.0 (603.5–1092.0); *p* = 0.009). In immunocompromised patients with sepsis or septic shock, the median presepsin levels were significantly higher than those of patients without sepsis or septic shock (1708.0 (851.5–3017.0) vs. 689.0 (510.0–862.5); *p* = 0.004]; 1946.0 (1125.5–3376.5) vs. 689.0 (510.0–862.5); *p* < 0.001)). Similarly, additional results showed a significant difference in median procalcitonin values among patients in the no sepsis, sepsis, and septic shock groups (no sepsis, 0.1 (0.1–0.4); sepsis, 5.7 (0.5–14.7); septic shock, 20.1 (5.9–59.0); *p* < 0.001), and among patients in the immunocompromised subgroup (no sepsis, 0.3 (0.1–0.5); sepsis, 3.6 (0.6–31.7); septic shock, 31.7 (5.7–59.0); *p* < 0.001).

Receiver operating characteristic (ROC) curves were generated to compare the presepsin and procalcitonin levels for diagnosing sepsis or septic shock (Figure 2 and Table 2). In all patients, presepsin showed an area under the curve (AUC) of 0.736 (95% CI, 0.637–0.834), with 50% sensitivity and 94.7% specificity, at cutoff point ≥ 1632; however, procalcitonin showed an AUC of 0.898 (95% CI, 0.819–0.976), with 87% sensitivity and 89.5% specificity, at cutoff point ≥ 0.64. According to this comparison, the AUC of procalcitonin was higher than that of presepsin (AUC 0.898 vs. 0.736, *p* = 0.007) (Figure 2A). In the immunocompromised subgroup, presepsin showed an AUC of 0.87 (95% CI, 0.765–0.975), with 66% sensitivity and 100% specificity, at cutoff point ≥ 1248. While procalcitonin showed an AUC of 0.892 (95% CI, 0.764–1), with 90% sensitivity and 87.5% specificity, in immunocompromised patients. The AUC between the two biomarkers was not significantly different (AUC 0.870 vs. 0.892, *p* = 0.766).

### 2.3. Correlations of Presepsin and Procalcitonin in Patients with Sepsis

Appendix A depicts correlations between the presepsin and procalcitonin levels on day 1. A very weak correlation was found between serum presepsin and procalcitonin levels in all patients (*r* = 0.037, *p* = 0.037), whereas no significant correlation was noted between those biomarkers in immunocompromised patients.

### 2.4. Presepsin Change and Mortality

Table 3 lists the demographic differences between survivors and non-survivors among 88 patients with sepsis. Twelve patients who died before 3 days after ICU admission were excluded from this analysis. A significantly larger proportion of immunocompromised patients was observed among non-survivors than among survivors. Unsurprisingly, non-survivors had a significantly higher SAPS 3 and SOFA score at ICU admission. Notably, presepsin levels on day 3, but not on day 1, were significantly higher in patients who died. Furthermore, the proportion of patients with ΔPresepsin+, which means plasma presepsin level on day 3 minus day 1 > 0, was significantly higher among non-survivors than among survivors; a similar result was observed in the subgroup of immunocompromised patients with sepsis (Appendix A). After adjusting for potential confounding factors, ΔPresepsin+ remained significantly associated with in-hospital mortality of immunocompromised patients with sepsis (adjusted OR, 6.22; 95% CI, 1.33–29.06; *p* = 0.02) (Table 4).

### 2.5. Subgroup Analysis for Non-Survivors

Figure 3 depicts the changes in presepsin and procalcitonin levels in non-survivors. The median presepsin levels on day 3 were significantly higher than those on day 1 in all patients who died in the hospital (1965.0 (1149.0–3423.0) vs. 1643.0 (777.0–3310.0); *p* = 0.046). In addition, in a subgroup of immunocompromised patients, the median presepsin levels on day 3 were significantly higher than those on day 1 (1449.0 (987.5–3594.0) vs. 1116.0 (773.0–3141.5); *p* = 0.018). There was no significant difference observed in procalcitonin levels between day 1 and day 3 in all patients or in the immunocompromised subgroup.

## 3. Discussion

This study aimed to evaluate the association between dynamic changes in presepsin levels and the prognosis of immunocompromised patients with sepsis. We found that the dynamic changes in presepsin levels between day 1 and day 3 were associated with a poor prognosis. The results in this study support the notion that assessing these changes in plasma presepsin levels could be a promising method to predict the prognosis of patients with sepsis. Presepsin can be combined with other clinical variables such as day 3 lactate or SAPS 3 score and may provide the prognostic granularity required for clinical use.

The most important finding of this study was that the dynamic changes in presepsin levels were associated with the prognosis of immunocompromised patients with sepsis. However, no significant association was observed between the changes in serum procalcitonin levels and poor prognosis. These results indicate that presepsin is useful for monitoring the prognosis of immunocompromised patients with sepsis as opposed to procalcitonin. Previous studies have shown that the changes in procalcitonin levels, according to the time course, were associated with in-hospital mortality [16,17]. However, serum procalcitonin levels were observed to decrease, even before recovering from sepsis, regardless of the prognosis [11,18]. The reason that the dynamic changes in presepsin levels differ from those of procalcitonin levels may be due to the different production mechanisms of each biomarker. The production of procalcitonin in patients with sepsis is induced by lipopolysaccharides (LPSs) and certain cytokines [19]. However, the increase in presepsin levels in patients with sepsis is associated with bacterial phagocytosis, independent of LPS and cytokines [20]. It has been shown that presepsin levels increased in a cecal ligation and puncture sepsis model; however, this was not observed in an LPS-induced sepsis model [21].

In this study, both presepsin and procalcitonin were able to distinguish patients with sepsis from those without sepsis. Although the diagnostic power of procalcitonin was better than that of presepsin, the diagnostic power of both biomarkers was not significantly different in immunocompromised patients. Baraka et al. also reported that the reliability of presepsin was similar to that of procalcitonin in predicting bacteremia in patients with febrile neutropenia [13]. These findings supported the rationale that presepsin is a useful biomarker in diagnosing sepsis in immunocompromised patients.

In contrast to earlier studies performed on non-immunocompromised patients in which presepsin levels on day 1 were significantly associated with a poor prognosis, in the present study, presepsin levels at ICU admission were not significantly associated with in-hospital mortality [8,9]. In this study, severity score and presepsin level at ICU admission and in-hospital mortality were significantly higher in immunocompromised patients than in immunocompetent patients. This difference might be explained by the effects of the underlying immunocompromised status and organ dysfunctions on mortality.

To the best of our knowledge, this is the first study that evaluated the role of dynamic changes in presepsin levels in predicting the prognosis of immunocompromised patients with sepsis. However, the present study had some limitations. First, the size of this study was relatively small. Second, the cause of immune suppression was heterogenous. Although this heterogeneity reflects real-world data, it might be possible that we missed findings that are specific to a particular subgroup. Third, the cause of sepsis was not included in this study. Therefore, studies with a large sample size and various pathological conditions are required to evaluate the prognostic power of presepsin in immunocompromised patients with sepsis.

## 4. Materials and Methods

### 4.1. Ethics Approval and Consent to Participate

This study was conducted in accordance with the relevant legislation and approved by the Ethics Committee of Seoul St. Mary’s Hospital, Seoul, Korea (KC18DESI0739) and complied with the Declaration of Helsinki and Good Clinical Practice guidelines. Informed consent was obtained from all participants included in the study.

### 4.2. Study Population

This study was conducted in Seoul St. Mary’s Hospital between March 2019 and June 2020. During this period, plasma samples were collected from patients who required ICU admission. Patients were classified into one of three diagnostic groups: no sepsis, sepsis, and septic shock. Sepsis and septic shock were diagnosed according to the third international consensus definitions for sepsis and septic shock (Sepsis-3) [1]. Immunocompromised patients were defined as patients with any of the following: HIV infection (all stages), neutropenia (neutrophil count < 1 × 109/L), exposure to glucocorticoids (>0.5 mg/kg for >30 days) and/or immunosuppressive or cytotoxic medications, solid organ transplantation, allogeneic or autologous stem cell transplantation, hematologic malignancy, or solid tumor [22]. On the day of ICU admission (day 1), the Sequential Organ Failure Assessment (SOFA) score and SAPS 3 were used as indicators of sepsis severity. We collected comorbidity data of the study participants and used them to calculate the Charlson comorbidity score index [23]. Variables shown in Table 1 were recorded at ICU admission and on day 3. Presepsin, procalcitonin, and lactate levels were measured in each blood sample.

The terms “ΔPresepsin” and “ΔProcalcitonin” were used to represent the day 3 minus day 1 plasma presepsin and procalcitonin levels, respectively. The terms “ΔPresepsin+” and “ΔProcalcitonin” were used for ΔPresepsin and ΔProcalcitonin values that were > 0, respectively.

This study was conducted in accordance with the relevant legislation and approved by the Ethics Committee of Seoul St. Mary’s Hospital (KC18DESI0739) and complied with the Declaration of Helsinki and Good Clinical Practice guidelines. Informed consent was obtained from all participants included in the study.

### 4.3. Presepsin Assay

Whole blood samples were collected using conventional blood collection tubes with heparin as an anticoagulant. The samples were centrifuged within 2 h after blood collection, following which the plasma was preserved at −80 °C until tested. Plasma presepsin levels were measured by a chemiluminescent enzyme immunoassay on the fully automated PATHFAST^®^ immunoanalyzer (LSI Medience Corporation, Tokyo, Japan).

### 4.4. Procalcitonin Assay

Serum procalcitonin levels were measured with fully automated chemiluminescent immunoassay using ADVIA Centaur B.R.A.H.M.S PCT (Siemens Healthcare Diagnostics, Berlin, Germany), according to the manufacturer’s instructions.

### 4.5. Statistical Analyses

All statistical analyses of our data were performed using the R 4.0.0 version (R Foundation, Vienna, Austria). The distribution of continuous variables was determined using the Shapiro–Wilk normality test. All results are reported as means ± standard deviations for normally distributed continuous variables and as medians and interquartile ranges (IQRs) for non-normally distributed continuous data. Categorical data are described as numbers and percentages. Patient characteristics were compared using the chi-squared test or Fisher’s exact test, as appropriate, for categorical variables, and independent samples t-tests for continuous variables. Receiver operating characteristic (ROC) curves were used to evaluate the ability of procalcitonin and presepsin to diagnose sepsis. DeLong’s test was used to compare the area under the ROC curves [24]. Wilcoxon rank–sum tests for paired data were used to assess presepsin levels between day 1 and day 3. Multivariate analysis was performed to investigate associations between patient characteristics and in-hospital mortality. Odds ratios (ORs) and the corresponding 95% confidence intervals (CIs) were computed. Goodness-of-fit was computed to assess the relevance of the logistic regression model. All statistical tests were two-tailed, and *p* < 0.05 was considered to be statistically significant.

## 5. Conclusions

In summary, we found that presepsin was a useful biomarker for diagnosing sepsis in immunocompromised patients. Furthermore, the current study showed that the dynamic changes in plasma presepsin levels between day 1 and day 3 were significantly associated with mortality in patients with sepsis.

## Figures and Tables

**Figure 1 diagnostics-11-00060-f001:**
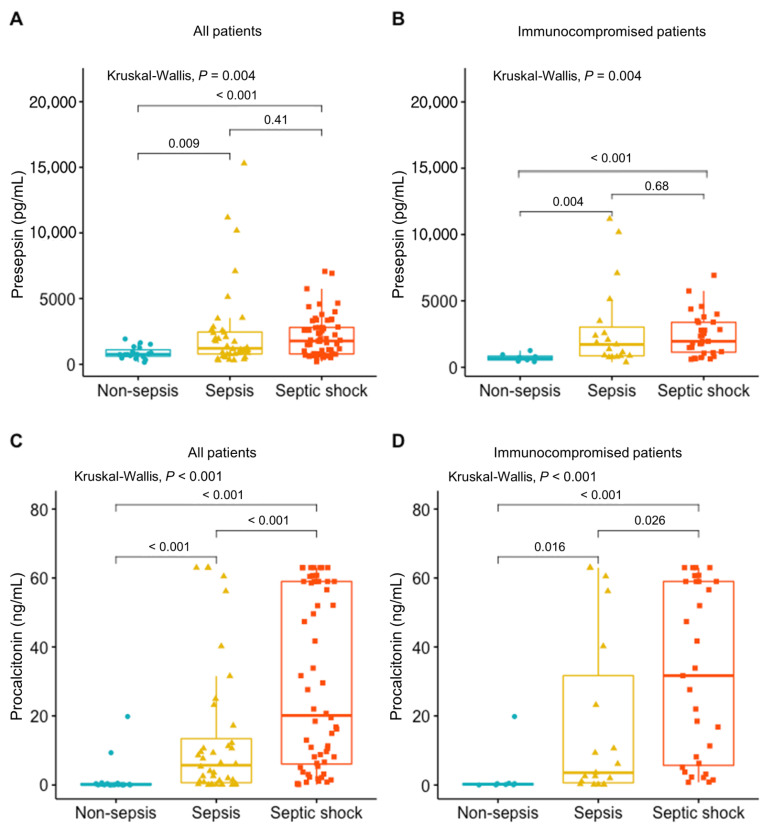
Comparison of presepsin value in different pathological conditions. (**A**) Presepsin value in all patients. (**B**) Presepsin value in immunocompromised patients. (**C**) Procalcitonin value in all patients. (**D**) Procalcitonin value in immunocompromised patients.

**Figure 2 diagnostics-11-00060-f002:**
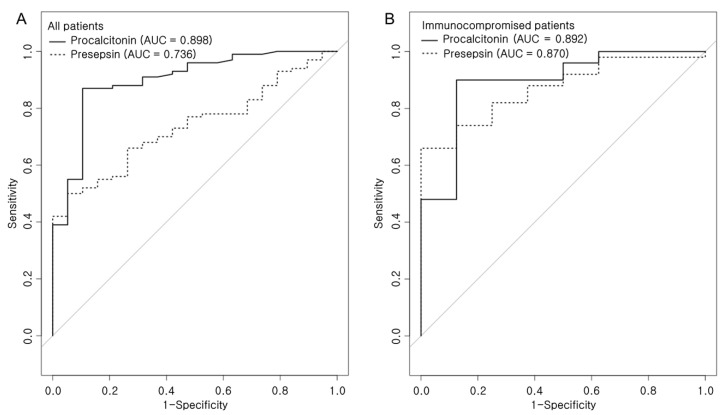
Pairwise comparison of receiver operating characteristic (ROC) curves of presepsin and procalcitonin level on the day of ICU admission (**A**) in all patients and (**B**) in immunocompromised patients.

**Figure 3 diagnostics-11-00060-f003:**
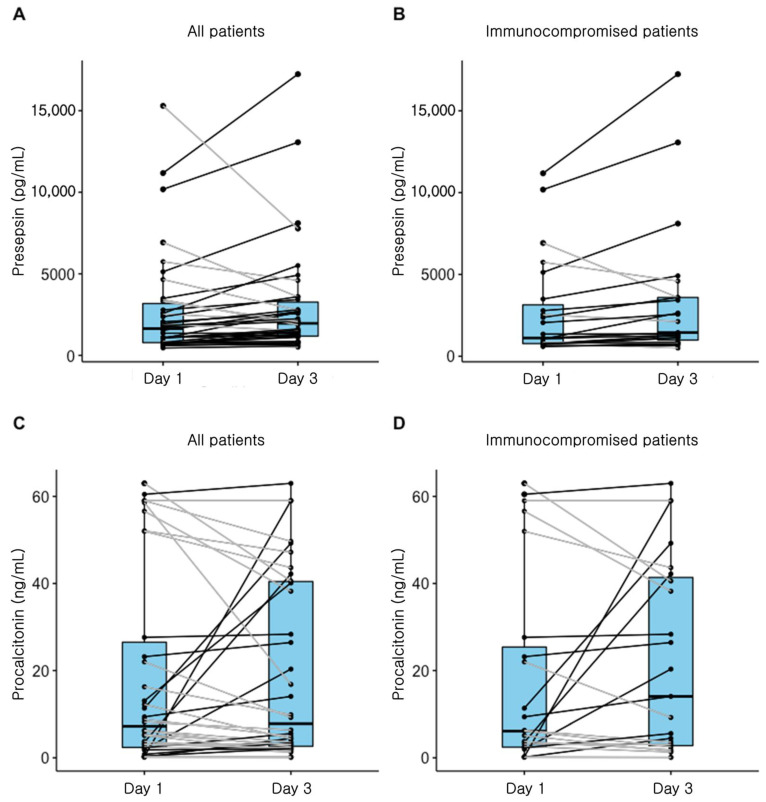
Change of presepsin and procalcitonin levels in non-survivors. (**A**) Presepsin levels in all patients with sepsis. (*p* = 0.046), (**B**) Presepsin levels in immunocompromised patients (*p* = 0.018), (**C**) Procalcitonin levels in all patients (*p* = 0.475), and (**D**) Procalcitonin levels in immunocompromised patients (*p* = 0.745).

**Table 1 diagnostics-11-00060-t001:** Comparison of baseline characteristics of the study population on intensive care unit (ICU) admission (N = 119).

	Total (n = 119)	Immunocompetent (n = 61)	Immunocompromised (n = 58)	*p* Value
Age, years	70.0 (58.0–79.5)	77.0 (68.0–83.0)	61.5 (51.0–70.0)	<0.001
Sex, male	62 (52.1)	34 (55.7)	28 (48.3)	0.528
Sepsis severity				
No sepsis	19 (16.0)	11 (18)	8 (13.8)	0.754
Sepsis	40 (33.6)	21 (34.4)	19 (32.8)	
Septic shock	60 (50.4)	29 (47.5)	31 (53.4)	
SOFA score	9.0 (6.5–11.5)	8.2 ± 4.1	9.9 ± 3.9	0.019
SAPS3 score	75.5 ± 14.9	71.7 ± 15.2	79.5 ± 13.5	0.004
Lactic acid	2.5 (1.3–4.8)	2.2 (1.3–4.1)	3.0 (1.5–7.5)	0.169
Presepsin	1254.0 (730.5–2569.5)	1065.0 (670.0–1935.0)	1734.5 (789.0–2831.0)	0.024
Procalcitonin	8.2 (0.6–32.8)	8.2 (0.3–19.5)	8.8 (0.9–56.2)	0.213
Charlson comorbidity index	5.0 (4.0–7.0)	5.0 (4.0–7.0)	5.0 (3.0–7.0)	0.509
Invasive ventilation on Day 1	53 (44.5)	27 (44.3)	26 (44.8)	1.000
Vasopressor on Day 1	77 (64.7)	38 (62.3)	39 (67.2)	0.710
Hospital mortality, (%)	53 (44.5)	19 (31.1)	34 (58.6)	0.005

SOFA, Sequential Organ Failure Assessment; SAPS3, Simplified Acute Physiology Score 3.

**Table 2 diagnostics-11-00060-t002:** Validity of presepsin and procalcitonin in diagnosis of sepsis or septic shock.

Variable	Cutoff	AUC	Sensitivity	Specificity	+ PV	− PV	*p* Value
In all patients							
Presepsin	1632	0.736	50.0	94.7	96.2	25.4	<0.001
Procalcitonin	0.64	0.898	87.0	89.5	96.7	55.2	<0.001
In immunocompromised patients							
Presepsin	1248	0.870	66.0	100	100	32.0	<0.001
Procalcitonin	0.5	0.892	90.0	87.5	97.8	58.3	<0.001

+ PV, positive predictive value; − PV, negative predictive value. AUC: area under the curve.

**Table 3 diagnostics-11-00060-t003:** Clinical and laboratory characteristics of survived and died patients with sepsis (N = 88).

	Survived Patient (n = 50)	Died Patient (n = 38)	*p* Value
Sex, male	29 (58.0)	18 (47.4)	0.439
Age, yrs	71.0 (61.0–79.0)	69.5 (54.0–83.0)	0.879
Immunocompromised patients	18 (36.0)	23 (60.5)	0.039
Charlson comorbidity index	6.0 (4.0–8.0)	5.0 (4.0–7.0)	0.589
SAPS3 score	72.2 ± 11.6	83.6 ± 12.9	<0.001
SOFA score	8.0 (7.0–11.0)	11.0 (8.0–13.0)	0.002
Use of vasopressor on Day 1	35 (70.0)	29 (76.3)	0.676
Use of invasive ventilation on Day 1	15 (30.0)	21 (55.3)	0.030
Systolic BP at ICU admission	79.0 (70.0–89.0)	82.0 (71.0–90.0)	0.787
Diastolic BP at ICU admission	47.5 (41.0–55.0)	50.0 (41.0–56.0)	0.385
Heart rate at ICU admission	124.2 ± 25.8	138.6 ± 28.2	0.014
Respiratory rate at ICU admission	28.0 (23.0–34.0)	31.5 (26.0–38.0)	0.076
Leukocyte count, Day 1	8.9 (4.3–17.3)	10.0 (3.0–16.5)	0.768
Neutrophil count, Day 1	7.1 (3.3–15.2)	8.6 (2.8–14.1)	0.714
Platelet count, Day 1	156.5 (52.0–235.0)	64.0 (34.0–143.0)	0.006
Lactate, Day 1	2.4 (1.3–3.9)	3.4 (1.9–7.1)	0.078
Lactate, Day 3	1.4 (1.0–2.2)	2.6 (1.6–5.8)	<0.001
Procalcitonin, Day 1	10.6 (1.5–49.6)	7.2 (2.3–27.6)	0.383
Procalcitonin, Day 3	3.5 (0.7–13.8)	7.8 (2.5–40.6)	0.022
ΔProcalcitonin+	4 (8.0)	14 (36.8)	0.002
Presepsin, Day 1	1209.0 (623.0–2559.0)	1643.0 (777.0–3310.0)	0.135
Presepsin, Day 3	933.0 (638.0–1571.0)	1965.0 (1149.0–3423.0)	0.001
ΔPresepsin+	16 (32.0)	28 (73.7)	<0.001

SAPS3, Simplified Acute Physiology Score 3; SOFA. Sequential Organ Failure Assessment; ΔProcalcitonin+, serum procalcitonin level on day 3 minus day 1 > 0; ΔPresepsin+, plasma presepsin level on day 3 minus day 1 > 0.

**Table 4 diagnostics-11-00060-t004:** Logistic regression analysis for the in-hospital mortality in immunocompromised patients.

	Univariate Analysis		Multivariate Analysis	
	OR (95% CI)	*p* Value	OR (95% CI)	*p* Value
SAPS3 score	1.04 (0.98–1.09)	0.212		
SOFA score	1.08 (0.87–1.34)	0.480		
Systolic BP at ICU admission	1.04 (0.99–1.10)	0.092		
Diastolic BP at ICU admission	1.05 (0.99–1.12)	0.076		
Heart rate at ICU admission	1.01 (0.98–1.03)	0.510		
Respiratory rate at ICU admission	1.01 (0.92–1.11)	0.800		
Procalcitonin, Day 1	0.98 (0.96–1.01)	0.140		
Procalcitonin, Day 3	1.02 (0.98–1.05)	0.331		
ΔProcalcitonin+	6.15 (1.14–33.20)	0.035	5.45 (0.72–41.17)	0.100
Presepsin, Day 1	1.00 (1.00–1.00)	0.661		
Presepsin, Day 3	1.00 (1.00–1.00)	0.313		
ΔPresepsin+	7.20 (1.79–29.01)	0.006	6.22 (1.33–29.06)	0.020

SAPS3, Simplified Acute Physiology Score 3; SOFA. Sequential Organ Failure Assessment; ICU, intensive care unit; ΔProcalcitonin+, serum procalcitonin level on day 3 minus day 1 > 0; ΔPresepsin+, plasma presepsin level on day 3 minus day 1 > 0.

## Data Availability

The data presented in this study are available on request from the corresponding author. The data are not publicly available due to the consent provided by participants on the use of confidential data.

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
