# Peer review of "The Association between Dynamic Changes in Serum Presepsin Levels and Mortality in Immunocompromised Patients with Sepsis: A Prospective Cohort Study"

_diagnostics, 2021, doi:10.3390/diagnostics11010060_

Round 1

Reviewer 1 Report

The submitted manuscript by J Lee et al. is an interesting prospective cohort study investigating the utility of monitoring presepsin levels in immunocompromised patients with sepsis. They conclude that change in presepsin levels between day 1 and 3 is a useful marker to help with clinical prognostication. They found that presepsin performs similarly to procalcitonin for the diagnosis of sepsis/septic shock, particularly in immunocompromised patients. Additionally, the authors report that increases in presepsin from day 1 to 3 may allow for clinical prognostication. This manuscript contributes important data to the diagnostic world regarding the value of presepsin in the evaluation of sepsis. 

Minor Comments

  1. In Figure 2, the ROC curve corresponding to which marker is not labeled. The reader assumes that black represents immunocompromised. Additionally, the labels are difficult to read and their size should be increased. Finally, I would recommend making either procalcitonin or presepsin dashed to avoid confusion in black and white.
  2. In both Figures 2 and 3 the authors have not included a heading that makes it clear to the reader that the figures on the left are all patients and the right represent immunocompromised only. Adding such a label would improve clarity.
  3. For Figure 3, it would be helpful for the authors to colorcode (or stylize the lines) to allow for differentiation of those patients with increasing or decreasing presepsin levels more easily. For example, lines for patients with increasing levels could be black and those for decreasing levels could be grey. This would better convey the message the authors are attempting to convey.
  4. For Figure 3, for clarity the authors should also include the median for days 1 and day 3 in all panels as a distinct marker or colored line in keeping with the text.
  5. In section 2.3, the meaning of delta-presepsin+ is not clearly conveyed. This should be clarified in the manuscript text as it is described in the legend for Table 3
  6. In the discussion, lines 167-168, the authors contend that these results support the use delta-presepsin+ as an “effective method” for prognostication in immunocompromised patients. This statement should be softened as 33.3% of patients who survived also had a rise in presepsin levels. The authors should acknowledge that based upon the results from this cohort, delta-presepsis+ alone does not have adequate capacity to allow for prognostication. It would be reasonable to state that it is a promising marker that could be combined with other clinical variables (i.e. day 3 lactate, SAPS3 score, etc), which may provide for prognostic granularity required for clinical use.

Author Response

1. In Figure 2, the ROC curve corresponding to which marker is not labeled. The reader assumes that black represents immunocompromised. Additionally, the labels are difficult to read and their size should be increased. Finally, I would recommend making either procalcitonin or presepsin dashed to avoid confusion in black and white.

Thanks for your detailed comments. We made a change in figure 2 following your recommendation.

2. In both Figures 2 and 3 the authors have not included a heading that makes it clear to the reader that the figures on the left are all patients and the right represent immunocompromised only. Adding such a label would improve clarity.

Following your opinion, we added label that makes it clear to the reader that the figures on the left are all patients and the right represent immunocompromised only.

3. For Figure 3, it would be helpful for the authors to colorcode (or stylize the lines) to allow for differentiation of those patients with increasing or decreasing presepsin levels more easily. For example, lines for patients with increasing levels could be black and those for decreasing levels could be grey. This would better convey the message the authors are attempting to convey.

Following your opinion, we changed the color code in Figure 3.

4. For Figure 3, for clarity the authors should also include the median for days 1 and day 3 in all panels as a distinct marker or colored line in keeping with the text.

Thank you for your detailed comments. We added the median for days 1 and day 3 in all panels of Figure 3.

5. In section 2.3, the meaning of delta-presepsin+ is not clearly conveyed. This should be clarified in the manuscript text as it is described in the legend for Table 3.

Thank you very much for the valuable comments. Although we already made a definition of delta-presepsin+ in method section, we added it in section 2.4 (line 150-151 in revised manuscript) again following your recommendation.

6. In the discussion, lines 167-168, the authors contend that these results support the use delta-presepsin+ as an “effective method” for prognostication in immunocompromised patients. This statement should be softened as 33.3% of patients who survived also had a rise in presepsin levels. The authors should acknowledge that based upon the results from this cohort, delta-presepsis+ alone does not have adequate capacity to allow for prognostication. It would be reasonable to state that it is a promising marker that could be combined with other clinical variables (i.e. day 3 lactate, SAPS3 score, etc), which may provide for prognostic granularity required for clinical use.

Thank you for your comment. We do agree your opinion. We softened the statement following your opinion in the discussion section as “The results in this study support the notion that assessing these changes in plasma presepsin levels could be a promising method to predict the prognosis of patients with sepsis. Presepsin can be combined with other clinical variables such as day 3 lactate or SAPS 3 score, and may provide prognostic granularity required for clinical use.” in line 215-218.

Reviewer 2 Report

In this study, Jongmin et al aimed to evaluate the role of presepsin as a diagnostic biomarker for sepsis. The authors demonstrated an association between serum presepsin levels and mortality in sepsis patients. Since sepsis is one of the leading causes of death in ICUs, early diagnosis of sepsis is critical and this study will advance the use of presepsin as a diagnostic marker for sepsis. However, I have the following comments.

1-The authors should place the second paragraph of the discussion in the introduction section and should expand this section clearly describing the role, signaling secretion module of presepsin.

2-The authors demonstrated that there is an increase in presepsin and procalcitonin levels in sepsis patients. It will be interesting to see the correlation between these two markers.

3-Is there a difference in levels of soluble CD14 and other pro-inflammatory markers such as TNF, IL-6, IL1b, etc between the different patient groups. Does the correlate with presepsin?

4-The authors claimed that dynamic change in presepsin levels is a marker for sepsis severity, whereas the data shows there is no difference between day1 and day3 levels of presepsin levels in immunocompromised patients.

5-The figures need to be properly denoted in the text.

Author Response

  1. The authors should place the second paragraph of the discussion in the introduction section and should expand this section clearly describing the role, signaling secretion module of presepsin.

Thank you for your comment. We do agree your opinion. We moved the second paragraph of the discussion in the introduction section and expand the section by describing the role, signaling secretion module of presepsin.

  1. The authors demonstrated that there is an increase in presepsin and procalcitonin levels in sepsis patients. It will be interesting to see the correlation between these two markers.

Thank you very much for the valuable comments. Following your opinion, we added the contents at the 2.3. section and supplementary figure 1.

  1. Is there a difference in levels of soluble CD14 and other pro-inflammatory markers such as TNF, IL-6, IL1b, etc between the different patient groups. Does the correlate with presepsin?

Thank you for your comments. However, we didn’t measure other pro-inflammatory markers such as TNF, IL-6, IL-1b, etc. We will compare presepsin with the inflammatory markers you mentioned in the next study.

  1. The authors claimed that dynamic change in presepsin levels is a marker for sepsis severity, whereas the data shows there is no difference between day1 and day3 levels of presepsin levels in immunocompromised patients.

Thank you for your detailed comments. In immunocompromised patients who died in the hospital, presepsin levels on day 3 were significantly higher than those on day 1. Furthermore, the proportion of patients with ΔPresepsin+, which means plasma presepsin level on day 3 minus day 1 > 0, was significantly higher among non-survivors than among survivors in the subgroup of immunocompromised patients with sepsis as shown Figure 3 and Supplementary Table 1. Based on these evidence, we claimed that dynamic change in presepsin levels could be a promising method to predict the prognosis of patients with sepsis.

  1. The figures need to be properly denoted in the text.

In response to your opinion, we revised and rearranged the figures. Thank you very much.
